# The Significance of Posterior Occlusal Support of Teeth and Removable Prostheses in Oral Functions and Standing Motion

**DOI:** 10.3390/ijerph18136776

**Published:** 2021-06-24

**Authors:** Kyosuke Oki, Yoichiro Ogino, Yuriko Takamoto, Mikio Imai, Yoko Takemura, Yasunori Ayukawa, Kiyoshi Koyano

**Affiliations:** 1Section of Fixed Prosthodontics, Division of Oral Rehabilitation, Faculty of Dental Science, Kyushu University, Fukuoka 812-8582, Japan; o-ki@dent.kyushu-u.ac.jp (K.O.); ayukawa@dent.kyushu-u.ac.jp (Y.A.); 2Department of Dentistry, School of Dentistry, Kyushu University, Fukuoka 812-8582, Japan; takamoto.yuriko.997@s.kyushu-u.ac.jp; 3Section of Implant and Rehabilitative Dentistry, Division of Oral Rehabilitation, Faculty of Dental Science, Kyushu University, Fukuoka 812-8582, Japan; mikio@dent.kyushu-u.ac.jp (M.I.); tomitay0804@dent.kyushu-u.ac.jp (Y.T.); 4Division of Advanced Dental Devices and Therapeutics, Faculty of Dental Science, Kyushu University, Fukuoka 812-8582, Japan; koyano@dent.kyushu-u.ac.jp

**Keywords:** posterior occlusal support, maximum occlusal force, masticatory function, standing motion, removable prostheses, Eichner index

## Abstract

The purpose of this study was to evaluate the effect of posterior occlusal support of natural teeth and artificial teeth on oral functions and standing motion. Patients who had been treated with removable prostheses were enrolled as the subjects. Their systemic conditions (body mass index (BMI) and skeletal muscle mass index (SMI)) were recorded. The subjects were classified into two groups according to a modified Eichner index: B1–3 (with posterior occlusal support) and B4C (without posterior occlusal support). Maximum occlusal force (MOF), masticatory performance (MP), and standing motion (sway and strength) were evaluated for cases with and without removable prostheses. There were no significant differences in BMI and SMI between the B1–3 group and the B4C group. The subjects with removable prostheses demonstrated significantly higher values in MOF, MP, and sway and strength than the subjects without removable prostheses. The comparison of oral functions between the B1–3 group and the B4C group revealed that the positive effect of posterior occlusal support of natural teeth and removable prostheses and the significant positive effects of posterior occlusal support on standing motion were partly observed in these comparisons. Posterior occlusal support of natural teeth and even of removable prostheses may contribute to the enhancement of oral functions and standing motion.

## 1. Introduction

One of the main causes of disability in the elderly is an accidental fall [1,2]. Falling is a severe problem for the elderly because it results in musculoskeletal injuries, brain injuries, and death in serious circumstances [3,4]. Multiple factors such as aging or aging-related physical dysfunctions, medication, cognitive impairment, and sensory deficits are well known as risk factors contributing to falls in the elderly [2,5,6,7]. Aging-related physical dysfunctions are inextricably associated with frailty and sarcopenia [8,9]. A decline in muscle mass and function due to the aging-related muscle atrophy is a characteristic feature of sarcopenia and is likely a cause of frailty [8,9,10,11]. One’s nutritional condition is closely related to muscle and bone aging, and good nutrition and physical exercise may be protective against frailty and sarcopenia [11,12,13,14].

Mastication and deglutition play crucial roles in nutritional management [15]. Healthy teeth, oral tissues including the tongue, and well-functioned prostheses are prerequisites for these functions. It has been reported that these functions are also impaired by aging and poor oral management, known as oral frailty, oral sarcopenia, and oral hypofunction [16,17,18,19,20]. The adverse effects of a decline of these functions on systemic and nutritional status have been reported [15,16,17,18,19,20,21,22]. Conversely, occlusal support is also important in oral functions, especially mastication [23,24,25,26]. This suggests that rehabilitation of occlusal support contributes to the prevention of frailty and sarcopenia indirectly. This suggestion may imply that rehabilitation of occlusal support has a positive effect on the prevention of accidental falls through the rehabilitation of mastication and nutritional status. Recent observational studies also demonstrated the association between occlusal support and physical function [27,28,29,30,31,32]. However, comparative studies that evaluate the effect of occlusal support and its rehabilitation on physical functions are still scarce.

The present study aimed to evaluate the effect of posterior occlusal support on standing motion and oral functions in the elderly. The subjects were categorized according to the presence or absence of posterior occlusal support of natural teeth and functional crowns (pontics), and oral functions and standing motion were compared between subjects with and without posterior occlusal support. In addition, the effects of posterior occlusal rehabilitation with removable prostheses on oral functions and standing motion were evaluated. The null hypotheses of this study were that there are no differences in oral functions and standing motion between subjects with and without occlusal support and between subjects with and without removable prostheses.

## 2. Materials and Methods

### 2.1. Ethical Approval

The protocol of the present study was developed in accordance with the ethical principles of the Declaration of Helsinki and was approved by our Institutional Review Board for Clinical Research (#2019–167). Patients who participated and provided written informed consent were enrolled as subjects.

### 2.2. Study Population

The patients who visited the Department of Prosthodontics, Kyushu University Hospital, between April 2019 and February 2020 were considered for enrollment as subjects of this study. The inclusion criteria of the present study were as follows: (1) patients who were more than 65 years old; (2) patients whose activities of daily living (ADL) were almost normal; and (3) patients who were rehabilitated with conventional removable prostheses by the Department of Prosthodontics, Kyushu University Hospital, and who could use their dentures without any specific problems. Thus, subjects were categorized into the groups Eichner B or C according to the original Eichner index. The exclusion criteria were as follows: (1) patients with systemic and/or localized diseases and medications that affect physical functions and with oral diseases that affect masticatory functions; (2) patients who could not understand the aim of this study due to cognitive impairment, etc.; and (3) patients with fixed prostheses supported by implants, implant-assisted partial removable dentures, or implant overdentures. As a result, this study’s subjects included 48 patients (21 males and 27 females, median age: 73, and interquartile range (IQR): 70–79).

The subjects were classified into two groups according to the Eichner index with our modifications [23,24,26]. The Eichner index was defined as follows: number of residual teeth was defined as the number of functional tooth crowns. Thus, pontics in fixed partial dentures were counted as residual teeth and remaining roots were excluded. Based on functional teeth and occlusal contacts, the subjects were classified into two groups. The first group included 24 subjects (11 males and 13 females, median age: 73, and IQR: 70–78) who had posterior occlusal support in the molar and/or premolar regions (Eichner B1, B2, and B3: B1–3 group). The second group included 24 subjects (10 males and 14 females, median age: 73, and IQR: 70–79) who had no posterior occlusal support (Eichner B4, C1, C2, and C3: B4C group).

### 2.3. Patient Profiles

In addition to age, gender, and the Eichner index, body mass index (BMI) for the degree of obesity (a risk factor for falls [33]) and skeletal muscle mass index (SMI) for the prevalence of sarcopenia [34,35] were calculated using the multi-frequency body composition meter (MC–780A, TANITA Corp., Tokyo, Japan) (Figure 1).

### 2.4. Measurements of Oral Function

#### 2.4.1. Maximum Occlusal Force (MOF)

The MOF was measured using a film for the occlusal force measurement system (Dental Prescale II and bite force analyzer, GC Co., Tokyo, Japan) (Figure 2) [18,20,23,24,26]. The subjects were asked to clench the film in the intercuspal position for 3 s. The clenched film was scanned by using the occlusal force analysis software to determine the MOF.

#### 2.4.2. Masticatory Performance (MP)

The MP was measured in the manner previous studies utilized to evaluate results [23,24,25,26]. In brief, the patients were instructed to voluntarily chew 2 g of gummy jelly for 20 s. The chewed gummy jelly was then moved to a cup with saliva and rinsing water, and the concentration of glucose dissolved in water was measured using a measuring device (Gluco Sensor GS–II, GC Co., Tokyo, Japan) (Figure 3).

### 2.5. Analyses of Standing Motion

To evaluate physical activity, muscle functions during the action of standing up were analyzed using a motor function analyzer (zaRitz BM–220, TANITA Corp., Tokyo, Japan) according to the manufacturer instructions. In brief, the subjects were asked to sit on a chair with their feet on the analyzer. The subjects stood up quickly, paused for 3 s, and sat down again. The subjects repeated this motion 3 times with and without their removable prostheses. The analyzer could evaluate sway and strength [36]. Sway is an index combining the degree of motion when standing up and the time until the shaking stops, while strength is an index combining leg muscle strength and standing speed (Figure 4).

### 2.6. Statistical Analyses

Numerical data were presented as the median and IQR and demonstrated as a box plot. The statistical analyses were conducted with the statistical package IBM SPSS Statistics 19 software (IBM Corp., Chicago, IL, USA).

The profiles of the subjects (age, BMI, and SMI) were statistically compared between the B1–3 group and the B4C group using the Mann–Whitney U test. To evaluate the effect of removable prostheses on oral functions and standing motion, the measurement items (MOF, MP, and sway and strength) in all subjects and in each group (B1–3 group and B4C group) were statistically compared between the values of those with and without their removable prostheses using the Wilcoxon signed-rank test. To evaluate the effect of posterior occlusal support on oral functions and standing motion, these measurement items were also compared between the B1–3 group and the B4C group using the Mann–Whitney U test. These comparisons were performed in the presence and absence of their removable prostheses. A value of less than 0.05 was considered statistically significant.

## 3. Results

### 3.1. Profiles of the Subjects

The profiles of the subjects are shown in Table 1 including the data of age and number of patients. There were no significant differences between the B1–3 group and the B4C group in all items (*p* > 0.05, Mann–Whitney U test). All subjects demonstrated normal ADL, although some patients were defined as underweight (one male and four females) or overweight (six males and nine females) according to the BMI results and the condition of sarcopenia (two males and five females) from the SMI results.

### 3.2. Comparisons of Oral Functions and Standing Motion in All Subjects with and without Removable Prostheses

The MOF and MP with and without removable prostheses were compared in all subjects (Figure 5). There were significant differences in both items between subjects with and without their removable prostheses (*p* < 0.01, Wilcoxon signed-rank test), indicating that rehabilitation of posterior occlusal support with removable prostheses could improve MOF and MP.

The results of standing motion analyses including strength and sway in all subjects with and without their removable prostheses are shown in Figure 6. There were significant differences in strength and sway between those with and without dentures (*p* < 0.05, Wilcoxon signed-rank test).

### 3.3. Comparisons of Oral Functions and Standing Motion between the B1–3 Group and the B4C Group with and without Removable Prostheses

The MOF and MP without removable prostheses were statistically compared between the B1–3 group and the B4C group. Compared with the B4C group, the B1–3 group exhibited significantly higher values in MOF and MP (*p* < 0.01, Mann–Whitney U test) (Figure 7). The MOF and MP with removable prostheses were also compared between the B1–3 group and the B4C group, and the subjects belonging to the B1–3 group exhibited statistically higher values in both functions compared to the subjects in the B4C group (*p* < 0.05, Mann–Whitney U test) (Figure 7). These findings suggest the significance of posterior occlusal support in both oral functions.

The results of the standing motion analyses with and without removable prostheses are shown in Figure 4. The subjects without removable prostheses in the B1–3 group exhibited significantly higher (better) values in sway than the subjects did without removable prostheses in the B4C group (*p* < 0.05, Mann–Whitney U test), although other comparisons (B1–3 vs. B4C in strength with and without removable prostheses, and B1–3 vs. B4C in sway with removable prostheses) did not detect significant differences (*p* > 0.05, Mann–Whitney U test) (Figure 8).

## 4. Discussion

It has been reported that poor oral health is closely associated with adverse health outcomes [16,20,21]. Malnutrition is attributed to poor oral status and function [15,22], resulting in sarcopenia and physical frailty [11,12,13,14]. Malnutrition has been considered the indirect effect of poor oral health on systemic condition. Several studies demonstrated the direct effect of oral functions on physical condition [37,38]. Above all, the effect of occlusal support on physical condition has been reported [27,28,29,30,31,32]. However, these studies were conducted as observational studies. The present study evaluated the effect of posterior occlusal support and rehabilitation with removable prostheses on standing motion in the elderly as a comparative study.

The BMI and SMI of the subjects in the present study are shown in Table 1. No significant differences in BMI and SMI between the B1–3 group and the B4C group were observed. All subjects demonstrated normal ADL, although some patients were defined as underweight or overweight and had sarcopenia. These factors might be confounding factors in this study. Comparisons based on the classification by BMI and SMI were not performed because of the limited number of subjects. Future studies that focus on both factors and occlusal support are advised to use more subjects.

Oral functions (MOF and MP) were enhanced in subjects with removable prostheses compared to subjects without removable prostheses (Figure 1 and Figure 3). In addition, the subjects with posterior occlusal support exhibited statistically higher MOF and MP values than the subjects without posterior occlusal support (*p* < 0.001, Mann–Whitney U test) and significant differences were detected when comparing both functions between both subjects with removable prostheses (*p* < 0.05, Mann–Whitney U test) (Figure 3). These findings clearly demonstrate that posterior occlusal support is strongly related to MOF and MP and that rehabilitation with removable prostheses contributes to the recovery or improvement of MOF and MP. The previous studies demonstrated similar results and provided more detailed data [23,24,25,26]. It is concluded that posterior occlusal support, even when reconstructed with removable prostheses, can play a crucial role in MOF and MP. Furthermore, based on the findings of the previous studies that demonstrated improvements in MOF and MP with removable prostheses [23,24,25,26], it is suggested that removable prostheses in this study works well in oral rehabilitation.

Our results confirmed that rehabilitation of posterior occlusal support with removable prostheses could improve standing motion (sway and strength) (Figure 2). Although the values measured by this device were novel and may lack scientific evidence, the measurement was very simple and an objective assessment considered it possible [36]. While we recognize the weak aspects of the measurements in this study, a significant difference of sway between the B1–3 group and the B4C group without removable prostheses suggests that posterior occlusal support is partly associated with standing motion (Figure 4). Furthermore, sway and strength with removable prostheses were statistically greater that those without removable prostheses, except for strength in the B1–3 group (Figure 4). These results also suggest that the rehabilitation of posterior occlusal support with removable prostheses can contribute to the improvement of standing motion; this effect was more striking in the B4C group in which subjects had no posterior occlusal support. Some discussions regarding the association between physical functions, especially balance (sway in this study), and posterior occlusal support have been reported [39,40]. These suggest plausible evidence for the masticatory and cervical muscles and that afferent signals from dental occlusion may be effective for balance control. These are related to the stability of the jaw position and occlusal support, and the rehabilitation with removable prostheses may also contribute to stability, resulting in improvements in standing motion. A previous study reported the contribution of occlusal support by artificial teeth to improve health and oral function [28]. The present study revealed that standing motion was improved by rehabilitation with removable prostheses and suggests an enhancement of physical functions, although further studies are required to elucidate this hypothesis.

The important issues in the present study are stated as follows. First, the subjects in this study were categorized based on their BMI and SMI, as described above, although they were all healthy and demonstrated normal ADL. It has been reported that these factors may be related to physical functions. Although our statistical analyses illustrated no significant differences between the B1–3 group and the B4C group, future studies will be expected to investigate the effects of occlusal support and these indexes on physical functions with more subjects. Second, there are various methods to assess physical functions [27,28,29,30,31]. The focuses of this study were oral function (MOF and MP) and standing motion (sway and strength), and the results revealed the effect of posterior occlusal support on a portion of physical functions. The background of falls was described previously; however, it is impossible to describe the effect of posterior occlusal support on the prevention of falls.

Lastly, there may be multiple confounding factors that affect the results of this study. The previous study mentioned systemic disease, medicine, and habits as potential confounding factors [31]. more subjects are required to investigate the association of factors such as BMI and SMI with physical function. In addition, the effect of rehabilitation with RPD or the strength of occlusal support with RPD may be different depending on teeth distribution (intermediate or free-end partial edentulism). Furthermore, the number of subjects in this study was limited, as mentioned in the inclusion and exclusion criteria, and it was difficult to calculate the sample size due to the lack of previous studies similar to the present study, unfortunately. However, we believe that this study demonstrated the positive effect of posterior occlusal support of natural teeth and removable prostheses on standing motion and suggests the importance of maintaining healthy teeth and encouraging prosthetic intervention from the viewpoint of physical function.

## 5. Conclusions

Prosthetic rehabilitation through removable prostheses could improve oral functions (MOF and MP) significantly. Moreover, the results of the present study clearly rejected our null hypotheses that there are no differences in standing motion between subjects with and without occlusal support of natural teeth and between subjects with and without removable denture rehabilitation. However, there are multiple confounding factors including BMI and SMI, and future studies with more subjects are necessary to classify the subjects based on these factors for further evaluations.

## Figures and Tables

**Figure 1 ijerph-18-06776-f001:**
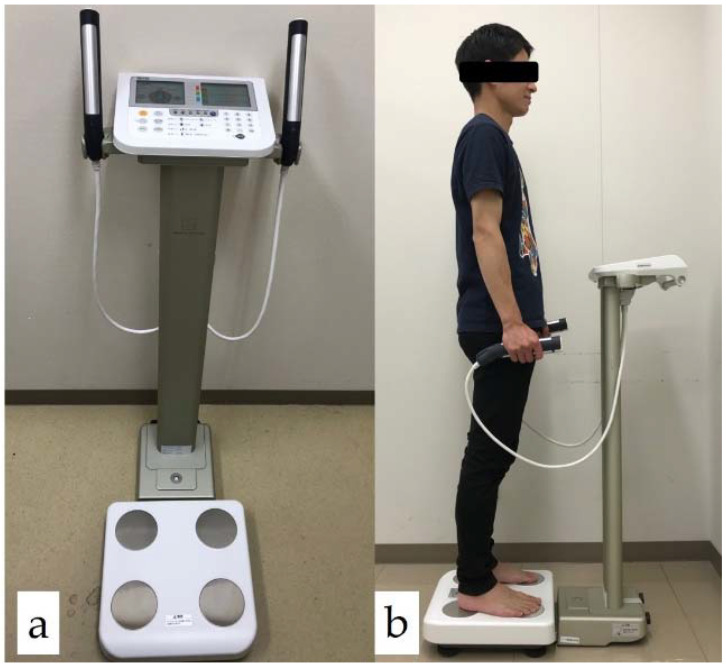
Measurement device (MC–780A) for the skeletal muscle mass index (SMI): (**a**) main unit; (**b**) measurement image.

**Figure 2 ijerph-18-06776-f002:**
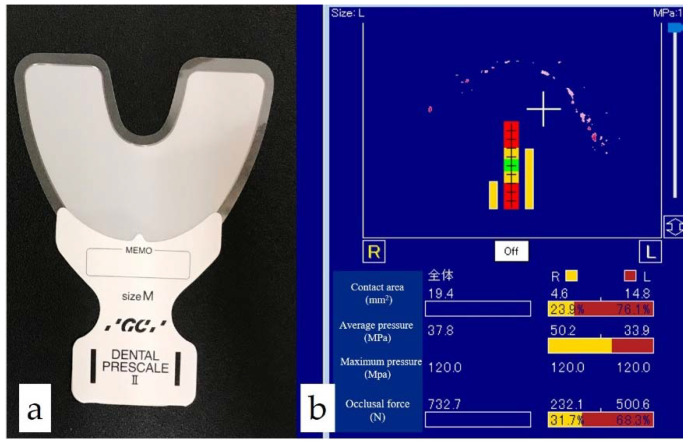
Occlusal force measurement system: (**a**) pressure-sensitive sheet (Dental Prescale II); (**b**) image of occlusal condition using software (bite force analyzer).

**Figure 3 ijerph-18-06776-f003:**
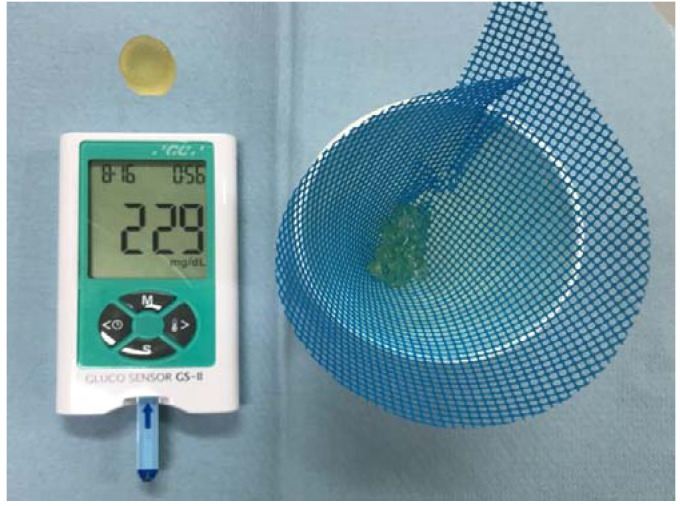
Masticatory performance (MP) measurement system (Gluco Sensor GS–II). The concentration of glucose from the chewed gummy jelly was defined as MP.

**Figure 4 ijerph-18-06776-f004:**
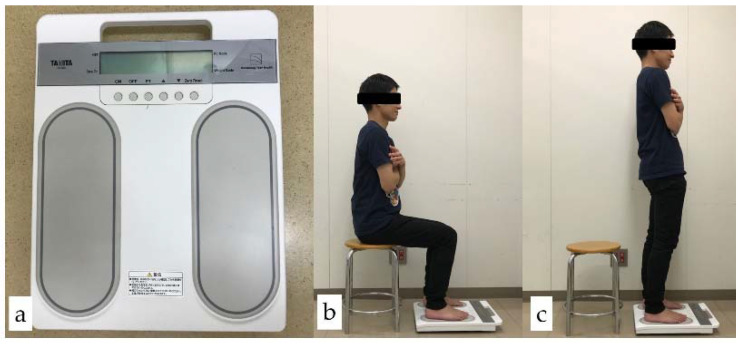
Motor function analyzer (zaRitz BM–220): (**a**) main unit; (**b**) measurement image (the beginning of analysis); (**c**) measurement image (motor function measurement after standing up).

**Figure 5 ijerph-18-06776-f005:**
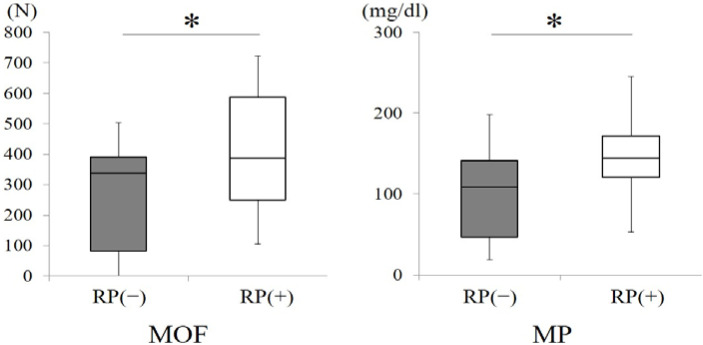
Comparisons of maximum occlusal force and masticatory performance between all subjects with and without removable prostheses. (* *p* < 0.01, Wilcoxon signed-rank test) RP (−)): subjects without removable prostheses; RP (+): subjects with removable prostheses.

**Figure 6 ijerph-18-06776-f006:**
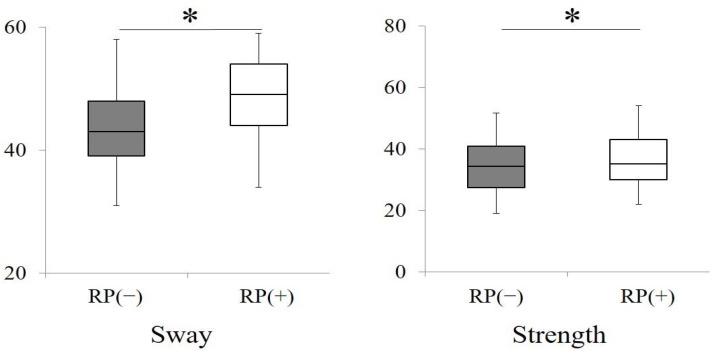
Comparisons of standing motion (sway and strength) between all subjects with and without removable prostheses. (* *p* < 0.05, Wilcoxon signed-rank test) RP (−): subjects without removable prostheses; RP (+): subjects with removable prostheses.

**Figure 7 ijerph-18-06776-f007:**
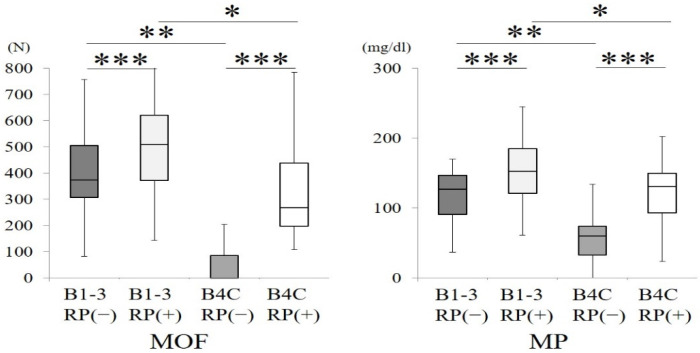
Comparisons of maximum occlusal force and masticatory performance between the B1–3 group and the B4C group with or without removable prostheses in each group (* *p* < 0.05, ** *p* < 0.01, Mann–Whitney U test). Comparisons of maximum occlusal force and masticatory performance between subjects with and without removable prostheses in each group (B1–3 group or B4C group) (*** *p* < 0.01, Wilcoxon signed-rank test). RP (−): subjects without removable prostheses; RP (+): subjects with removable prostheses.

**Figure 8 ijerph-18-06776-f008:**
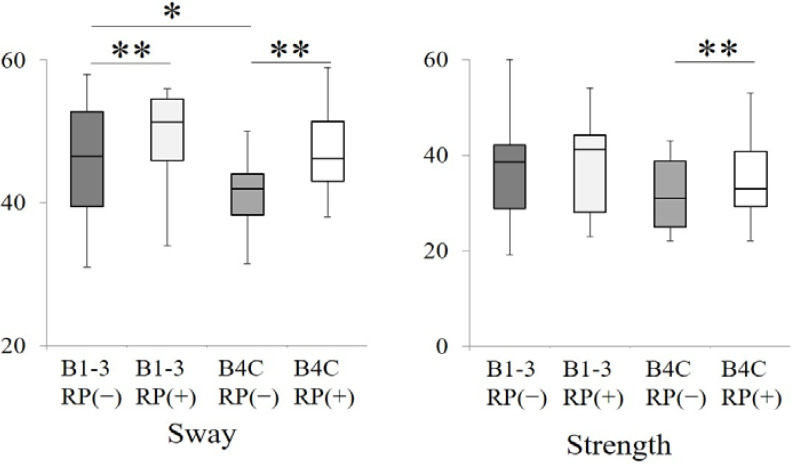
Comparisons of standing motion (sway and strength) between the B1–3 group and the B4C group with or without removable prostheses in each group (* *p* < 0.05, Mann–Whitney U test). Comparisons of standing motion (sway and strength) between subjects with and without removable prostheses in each group (B1–3 group or B4C group) (** *p* < 0.05, Wilcoxon signed-rank test). RP (+): subjects with removable prostheses; RP (−): subjects without removable prostheses.

**Table 1 ijerph-18-06776-t001:** Summary of subjects’ profiles; IQR: interquartile range; BMI: body mass index; SMI: skeletal muscle mass index. Statistical analyses: B1–3 group vs. B4C group (*p* > 0.05, Mann–Whitney U test in all items).

		All Subjects	B1–3 Group	B4C Group
		n = 48	n = 24	n = 24
Age (median and IQR)		73 (70–79)	73 (70–78)	73 (70–79)
Gender (male and female)		21:27	11:13	10:14
BMI (median and IQR)	All	23.1 (20.8–26.4)	22.4 (20.8–25.2)	23.5 (20.6–26.4)
	Male	23.5 (21.5–25.2)	23.2 (21.5–24.7)	24.1 (21.6–25.9)
	Female	22.3 (20.4–26.7)	22.2 (19–26.19)	23.1 (20.6–26.9)
SMI (median and IQR)	All	6.6 (6.1–7.7)	6.5 (6–7.1)	6.6 (6.1–7.7)
	Male	7.8 (7.0–8.5)	7.8 (7.4–8.6)	7.6 (6.6–8.4)
	Female	6.4 (5.9–7.0)	6.5 (5.9–7.2)	6.3 (5.8–6.6)

## Data Availability

The datasets used and/or analyzed during the current study are available from the corresponding author upon reasonable request.

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
