# Peer review of "The Significance of Posterior Occlusal Support of Teeth and Removable Prostheses in Oral Functions and Standing Motion"

_ijerph, 2021, doi:10.3390/ijerph18136776_

Round 1

Reviewer 1 Report

In general, the article shows us a correct Material and Method and a presentation on an unusual subject.
Perhaps, some images (patient, mouth, film for occlusal measuring system, motor function analyzer...) can help to understand the method.

Author Response

In general, the article shows us a correct Material and Method and a presentation on an unusual subject.
Perhaps, some images (patient, mouth, film for occlusal measuring system, motor function analyzer...) can help to understand the method.

>We appreciate your comments. As you pointed, we added some images (figures) in the text.

Figure 1 Multi frequency number body composition meter for SMI measurement

Figure 2 Occlusal measurement system for MOF measurement: (a) A pressure-sensitive sheet (Dental Prescale â…¡); (b) Image of Bite force analyzer

Figure 3 MP measurement device (Gluco Sensor GS–II)

Figure 4 Motor function analyzer (zaRitz BM–220)

Reviewer 2 Report

This is a novel perspective and comprehensive study about the holistic benefit of posterior occlusal support. Important heath implication supported by functional occlusion and rehabilitation. This manuscript definitely deserves publication. Some minot comments as follows: 1. Are all the RPD well-fitting? Could the authors comment on posterior support provided by ill-fitting RPD? 2. Maybe it is good to clarify to general readers about the definition of loss of post support. And is there a difference for patients. with short arch (premolar to premolar) and unilateral loss. 3. Why is the benefits on sway seems more significant than strengthen. Or what are the numbers entail? Could it considered clinically significant? Please comment on it.

Author Response

This is a novel perspective and comprehensive study about the holistic benefit of posterior occlusal support. Important heath implication supported by functional occlusion and rehabilitation.
This manuscript definitely deserves publication. Some minot comments as follows:

> Thank you for your comments. We’re proud of your response.

  1. Are all the RPD well-fitting? Could the authors comment on posterior support provided by ill-fitting RPD?

> As we mentioned in the text, the patients who could be registered in this study had to use their RPD without any specific problems. We were sure that PRD could work with well-fitting.

  1. Maybe it is good to clarify to general readers about the definition of loss of post support. And is there a difference for patients. with short arch (premolar to premolar) and unilateral loss.

>We believe you advised us to mention Eichner index. We’re afraid that a detailed explanation of Eichner index is complicated and we decided to cite previous studies [23,24,26] that showed this index in “2.2. Study population”. We hope it will be accepted. In addition, the number of subjects was limited and we gave up the detailed analysis regarding the distribution of residual teeth and the lactation of occlusal support. This could be a limitation of this study and we added some comments in “Discussion”.

  1. Why is the benefits on sway seems more significant than strengthen. Or what are the numbers entail? Could it considered clinically significant? Please comment on it.

>Actually, this is very difficult because no study evaluated the association between the measurement values and actual physical function (not standing motion). We understand that we need to prove the significance of these measurement value. However, the measurement values could be used as indicators to compare the physical functions in this study. We hope this report can inform the readers of this tool as one of methods for physical function evaluation. We’re doing a research that evaluates the association between these measurement values and the history of fall or some accidental events. We’ll report the result in future study. Some comments were added in “Discussion”.

Reviewer 3 Report

Thank you for giving me this opportunity to review the article entitled, " The significance of posterior occlusal support by teeth and removable prostheses in oral functions and standing motion".

1) Introduction - I think the introduction section should be improved, the text is very general and may present information that is not necessary for the interpretation of the article.

2) Materials and Methods: Inclusion and exclusion criteria need to be more detailed. Since this is an elderly population (over 65 years old), it is strange that there are no systemic diseases associated with any subject (as well as medication). It is also necessary to describe how the collection and selection of patients was carried out, as well as better explaining line 80 "oral diseases that affect masticatory function".

3)  The sample size was calculated based on the available literature or was a sample calculation tool used?

4) Please add the SPSS version. 

5) Table 1: please add the abbreviation subtitles;

6) Figure 4 is missing. 

Author Response

Thank you for giving me this opportunity to review the article entitled, " The significance of posterior occlusal support by teeth and removable prostheses in oral functions and standing motion".

>We’re glad to see your comments. We appreciate your support. The revisions of our article were as follows.

1) Introduction - I think the introduction section should be improved, the text is very general and may present information that is not necessary for the interpretation of the article.

>Thank you for your comment. We intended the structure as follows.

1st paragraph: the descriptions about physical dysfunction in the elderly because of aging, aging-related factors including sarcopenia. These descriptions were necessary because the subjects in this study had to be elderly and we evaluated SMI (sarcopenia). The previous studies indicated the association of sarcopenia with nutritional condition. Nutritional condition is related to food intake from the mouth, so this paragraph suggested the importance of oral (occlusal) condition and this was followed by 2nd paragraph.

2nd paragraph: the significance of oral condition including remaining teeth, occlusal support and some oral functions for nutritional management was described. In addition, the positive effect of occlusal support on physical function must be described because this is the main purpose of this study.

3rd paragraph: the purpose of this study.

We hope you can understand our intention and accept this structure of “Introduction”.

2) Materials and Methods: Inclusion and exclusion criteria need to be more detailed. Since this is an elderly population (over 65 years old), it is strange that there are no systemic diseases associated with any subject (as well as medication). It is also necessary to describe how the collection and selection of patients was carried out, as well as better explaining line 80 "oral diseases that affect masticatory function".

>We can agree your suggestions. Regarding first point, we excluded the patients whose systemic and/or localized disease and some medications that affect their physical function. We added some comments in this paragraph.

The subjects were limited to the patients who were treated by authors for qualitative consistency. And the study period was between April in 2019 and February in 2020 due to pandemic shut down of hospital. Unfortunately, we could not find the suitable previous study to calculate the sample size. Now, we’re conducting this study and in future study, we can calculate the sample size. We hope the reviewer understand the limitation of this study. Some comments were added including the response of 3) suggestion.

3)  The sample size was calculated based on the available literature or was a sample calculation tool used?

>As you can find above, we could not calculate the sample size because of the lack of previous study. In a future study, we believe that we can describe clearly.

4) Please add the SPSS version. 

>As you suggested, we revised.

5) Table 1: please add the abbreviation subtitles;

>As you suggested, we revised.

6) Figure 4 is missing. 

>I’m afraid that we could confirm Figure 4 in the text. However, Figure 4 was changed into Figure 8 because of the addition of some figures. We hope that you can find Figure 8 in the revised version.

Round 2

Reviewer 3 Report

The alterations made were in line with what was proposed.